# “Meaning in Life” Mediates the Relationship between Loneliness and Depressive Symptoms among Older Adults during the COVID-19 Pandemic

**DOI:** 10.3390/healthcare12050584

**Published:** 2024-03-04

**Authors:** Chanhee Kim, Jiyoung Chun

**Affiliations:** 1Department of Nursing, Changwon National University, 20 Changwondaehak-ro, Uichang-gu, Changwon-si 51140, Gyeongsangnam-do, Republic of Korea; chany131@changwon.ac.kr; 2College of Nursing, Sahmyook University, Seoul 01795, Republic of Korea

**Keywords:** loneliness, meaning in life, depression, older adults

## Abstract

Background: Loneliness was found to be a commonly experienced feeling among older adults during the COVID-19 pandemic and is considered to be a high-risk factor for depressive symptoms. Maintaining meaning in life has been found to be a protective resource for mental health among older adults. The purpose of this study was to examine the mediating effects of an individual’s meaning in life in the relationship between loneliness and depressive symptoms among older Korean adults. Methods: The data were obtained from a sample of 213 community-dwelling older adults aged over 65. The instruments were the UCLA Loneliness Scale, the Center for Epidemiologic Studies Depression Scale, and the Meaning in Life Questionnaire. Results: Loneliness was positively associated with depressive symptoms and negatively associated with the presence of and search for meaning among older adults. The presence of meaning mediated the relationship between loneliness and depressive symptoms but the search for meaning did not. The search for meaning mediated the relationship between loneliness and depressive symptoms through the presence of meaning. Conclusions: Our study findings suggest that efforts to reduce loneliness and improve meaning in life should be undertaken to prevent depressive symptoms among older adults.

## 1. Introduction

The global population is experiencing an unprecedented acceleration in aging, with the proportion of individuals aged 60 years and older expected to surge from 12% in 2015 to 22% in 2050 [1]. Furthermore, South Korea is transitioning into a super-aged society, with the proportion of individuals aged 65 years and over projected to exceed 40% by 2050, up from 17.5% in 2022 [2].

Depression is among the most prevalent mental disorders in later life, with reported prevalence rates in older South Korean adults ranging from 26.0% to 46.3% [3,4]. Geriatric depression has a complex impact on the physical, mental, and social well-being of older adults, including increased risk of suicide, cognitive decline, changes in sleep and appetite, social impairment, worsening of illness, and increased disability and mortality [5,6]. Thus, preventing geriatric depression is crucial to maintain older adults’ physical and mental health, daily functioning, social engagement, and overall quality of life. 

Older adults frequently encounter loneliness, which is a significant predictor of depression [7,8]. Loneliness and social isolation, often characterized as a global epidemic, affect approximately one-third of the older adult population worldwide, with 5% reporting frequent or constant feelings of loneliness [9,10,11]. In South Korea, the proportion of older adults living alone has been increasing annually from 18.8% of the total older adult population in 2016 to 20.8% in 2022 [2]. This increase can be attributed to factors such as longer life expectancy and adult children moving away from their parents. Additionally, the loss of a spouse through death leaves more older adults living alone in their old age. Moreover, owing to the COVID-19 pandemic, stay-at-home policies have led to decreased social interactions and intimate relationships among older adults [12,13], thus, contributing to increased social isolation and loneliness. Loneliness can have long-term negative outcomes, both physically and mentally [7]. When older adults experience loneliness, their health deteriorates, leading to visual and hearing impairments [10], and an increased risk of heart disease, depression, anxiety, dementia, cognitive decline, and even mortality rates [14,15]. In particular, it has been well established that an increased level of loneliness is closely related to the development of depression among older adults [16,17,18,19].

Having a sense of meaning or purpose in life is indicative of older adults’ overall well-being [20]. The pursuit of meaning in life is considered a fundamental human need and is the driving force behind human existence [21]. While finding meaning in life is important at any age, it holds particular significance for older adults, in the aspect of discovering the meaning of life as a unique existence based on their past life experiences. It encompasses the realization of the freedom and responsibility to make choices about one’s existence, which can aid in successfully concluding their life [20]. The existing literature has revealed that lower levels of meaning in life are associated with higher levels of depression [22,23,24,25]. Conversely, higher levels of meaning in life are associated with happiness, positive affect, well-being, and life satisfaction [26]. Meaning in life has been reported to alleviate difficulties and suffering in the face of life changes and crises later in life, such as illnesses, changes in living conditions, physical and cognitive decline due to aging, and loss [20,27]. Additionally, previous research has reported that meaning in life mediates the relationship between loneliness, psychological distress, and negative mental health outcomes [28,29]. Building on prior research, we anticipated that meaning in life would serve as a mediator in the relationship between loneliness and depressive symptoms among older adults.

Meaning in life has been conceptualized along two dimensions: the search for meaning, which pertains to the motivational aspect of seeking meaning in one’s life, and the presence of meaning, which involves how much meaning in life people experience [30,31]. There has been considerable research interest in understanding the respective functional roles of the search for and presence of meaning [32]. Therefore, it would be worthwhile to examine the mediating effects of the search for and presence of meaning on the relationship between loneliness and depression among older adults. 

This study examined the relationship between loneliness and depressive symptoms, whether this relationship can be mediated by meaning in life, and the differential role of the search for and presence of meaning in relation to depressive symptoms among older adults. Specifically, this study hypothesized that loneliness is related to greater depressive symptoms (H1); the two dimensions of meaning in life (search for and presence of meaning) mediate the relationship between loneliness and depressive symptoms (H2, H3); and the search for and presence of meaning have a serial mediating effect on the relationship between loneliness and depressive symptoms among older Korean adults (H4) (Figure 1).

## 2. Methods

### 2.1. Participants and Procedures

Data were collected from July to August 2021, during the COVID-19 pandemic. The eligibility criteria for study participation were as follows: (1) older adults living in their own homes or in non-institutional settings, (2) aged 65 years or older, and (3) able to communicate freely and offer written informed consent. Using a convenient sampling method, the participants were recruited from senior centers in Busan and Daegu, South Korea. After explaining the purpose of the study, a questionnaire was administered to participants who voluntarily agreed to participate. Data were collected by trained researchers using pen and paper. Trained researchers supported participants who asked for help to read the questionnaire. Participants took about 15 min to complete the questionnaires. This study aimed to have a sample size greater than the generally advised minimum of 200 for structural equation modeling (SEM) [32]. A total of 244 older adults were recruited and data from 213 were used for the final analysis because of incomplete responses from the remaining participants. This study was approved by the Institutional Review Board of the D. University (IRB No. 2-1040709-AB-N-01-202105-HR-031-04). This study was conducted in accordance with the Declaration of Helsinki and the guidelines for COVID-19 infection prevention. 

### 2.2. Measures

Data were collected on the participants’ general characteristics, including age, sex, education level, marital status, religion, and perceived health status. Perceived health status was assessed using three items: (1) What do you think about your current health status? (2) How does your current health status compare to a year ago? And (3), compared to others of the same age, how would you rate your health status? Responses to the above items were evaluated using a 5-point Likert scale, ranging from 1 (very unhealthy) to 5 (very healthy). Loneliness, depression, and meaning in life were measured using the following scales: 

The revised UCLA Loneliness Scale was used to assess loneliness in older adults. This scale was developed and revised by Russell and Peplau [33] and was validated using the Korean version by Kim [34]. This scale comprises 20 items rated on a four-point Likert scale ranging from 1 (never) to 4 (often). Higher scores indicate greater loneliness. These items showed good reliability in this study (Cronbach’s alpha = 0.89).

The Center for Epidemiological Studies Depression (CESD-10) Scale [35], a short version of the original 20-item version of the CES-D [36], was used to assess depressive symptoms among older adults. The validity of the Korean version of the CESD-10 scale has been well established [37]. This scale comprises 10 items rated on a four-point Likert scale, with higher scores indicating greater depressive symptoms. In this study, the Cronbach’s alpha for the revised CESD-10 was 0.80, indicating good reliability.

The Meaning in Life Questionnaire (MLQ), developed by Steger and Frazier [30], was used to assess meaning in life among older adults. The Korean version of the MLQ was validated by Won and Kim [38]. This scale comprises two subscales—the search for and presence of meaning in life—comprising five items. The presence of meaning subscale assesses participants’ perceptions of how meaningful their lives are, whereas the search for meaning subscale gauges their motivation to find meaning. The MLQ was rated on a seven-point Likert scale ranging from 1 (absolutely untrue) to 7 (absolutely true). Higher scores reflect a heightened sense of meaning in life. In this study, Cronbach’s alpha for the presence of meaning and search for meaning were 0.89 and 0.77, showing good reliability.

### 2.3. Analysis

Descriptive statistics were used to analyze the participants’ general characteristics and study variables. Pearson’s correlations were used to examine the relationships between the study variables. A structural equation model (SEM) was used to test the mediating effect. Maximum likelihood estimation was used to test the structural model. Multiple fit indices were used to evaluate the model. These include Chi-square (χ^2^), degrees of freedom (df), Tucker–Lewis index (TLI), comparative fit index (CFI), standardized root mean square residual (SRMR), and root mean square error of approximation (RMSEA). Goodness of fit was evaluated per the following criteria: χ^2^/df ≤ 3.00, TLI and CFI ≥ 0.90, SRMR, and RMSEA ≤ 0.08 [39]. Bootstrapping tests of mediation were conducted to examine whether meaning in life (search for and presence of meaning) mediated the relationship between loneliness and depressive symptoms. A total of 10,000 re-samplings were conducted, and 95% confidence intervals (CIs) were adopted. CIs that did not include zero indicated significance at 0.05. Further, IBM SPSS Statistics 22.0 and AMOS 22 were used for data analysis. 

## 3. Results

As shown in Table 1, participants’ ages ranged from 65 to 92 years (M = 75.07 years; SD = 6.60). Among the participants, 60.1% were women, and over half (60.1%) had a spouse. There were significant differences in depressive symptoms for the following variables: age, sex, having a spouse, education level, and perceived health status. Therefore, these were controlled as covariates in the SEM. 

Descriptive statistics are presented in Table 2. The skewness levels of the study variables were all less than two, and the kurtosis levels were all less than seven, indicating that they were normally distributed [40]. As shown in Table 2, loneliness was positively correlated with depressive symptoms (r = 0.53, *p* < 0.001). There were significant but weak negative correlations between loneliness and the search for meaning (r = −0.24, *p* < 0.001), as well as between loneliness and the presence of meaning (r = −0.41, *p* < 0.001). Additionally, significant but weak negative correlations were observed between the search for meaning and depressive symptoms (r = −0.18, *p* = 0.009), and between the presence of meaning and depressive symptoms (r = −0.36, *p* < 0.001). 

The SEM analysis results show that all fit indices were within an acceptable range: χ^2^ = 6.189, df = 3, χ^2^/df = 2.06, TLI = 0. 93, CFI = 0.99, SRMR = 0.03, and RMSEA = 0.07. As shown in Figure 2, loneliness negatively affected the search for meaning (β = −0.262, *p* < 0.001) and presence of meaning (β = −0.268, *p* < 0.001), and positively affected depressive symptoms (β = 0.479, *p* < 0.001). The search for meaning positively affected presence of meaning (β = 0.605, *p* < 0.001), but not depressive symptoms (β = 0.104, *p* = 0.915). The presence of meaning positively affected depressive symptoms (β = −0.159, *p* = 0.049). 

A bootstrap analysis was performed and the confidence intervals were estimated to examine the direct and indirect effects of H1, H2, H3, and H4. As shown in Table 3, the direct effect of loneliness on depressive symptoms was positively significant (β = 0.520, 95% CI = [0.413; 0.627]). The mediating effect of the presence of meaning on the relationship between loneliness and depressive symptoms was significant (β = 0.043, 95% CI = [0.003; 0.095]), but that of the search for meaning was not significant (β = −0.027, 95% CI = [−0.079; 0.008]. Moreover, the serial mediating effects of the search for meaning and presence of meaning between loneliness and depressive symptoms were significant (β = 0.025, 95% CI = [0.003; 0.063]. Based on the analyses, H1, H3, and H4 were supported, but H2 was not.

## 4. Discussion

This study examined the effect of loneliness on depressive symptoms and the serial mediating roles of the search for and presence of meaning in the relationship between loneliness and depressive symptoms among older Korean adults. 

As anticipated, our findings showed that loneliness is positively associated with depressive symptoms among older adults. Thus, H1 was supported. Our results are consistent with those of prior studies [41,42] that showed that loneliness predicted depressive symptoms during the COVID-19 pandemic. Findings from a meta-analysis and systematic review of 10 longitudinal studies show that loneliness is a critical risk factor for depressive symptoms [7,43]. During the pandemic, restrictions on social networks and outdoor activities led to an increased loneliness, which has affected depressive symptoms among older adults. Therefore, considering the increased levels of loneliness among older adults [44], our findings on the negative effects of loneliness on depression in older adults should be considered. 

The search for meaning had no mediating effect on the relationship between loneliness and depressive symptoms among older adults; thus, H2 was not supported. This finding was inconsistent with the assumptions of this study. Although the search for meaning could serve as a motivation for the discovery of meaning [21], it does not directly lead to decreased depressive symptoms. In previous studies, the relationships between the search for meaning and mental health have shown mixed findings. Empirical evidence has demonstrated that some studies have found a significant negative correlation between the search for meaning and depressive symptoms [45,46], whereas others have reported a significant positive correlation between these two factors [24,47,48]. These inconsistent findings can be attributed to the pursuit of meaning in life, which can yield tension [49] and be distressing; thus, the search for meaning may not necessarily lead to improved mental health. While positive mental health may not manifest immediately, engaging in this pursuit may expand the prospects for experiencing a sense of meaning and its subsequent effects.

Our findings also revealed that loneliness is associated with depressive symptoms through the presence of meaning in life among older adults. Thus, H3 was supported. Those with high loneliness reported a reduced presence of meaning in life, which was linked to increased depression. In the first half of the path, loneliness was significantly negatively associated with the presence of meaning among older adults, which is consistent with previous findings [31,50,51]. In the second half of the mediating path, the presence of meaning is significantly associated with depressive symptoms, which is supported by previous research [24,47,52]. This is also in line with prior studies that have found that loss of meaning has been consistently reported as leading to negative psychological outcomes, including suicidal ideation and behavior [53,54,55,56,57] and anxiety [22,24,47]. However, a sense of meaning in life is a beneficial psychological resource that promotes psychological well-being [32,52,58,59]. Individuals who perceive meaning and purpose are more likely to feel pleasant emotions; therefore, they may experience fewer negative emotions [60]. This also relates to Victor Frankl’s attitudinal pathway, which refers to the ways in which individuals view difficulties and suffering in their lives [49]. By employing a positive attitude when facing negative situations and hardships, individuals can discover meaning [49,61]; consequently, they are less prone to experiencing depressive symptoms. 

The mediating effect of the presence of meaning on the relationship between loneliness and depression indicates that the perception of meaning in life in lonely older adults may suppress depression. Prior studies have demonstrated the mediating effect of the presence of meaning. Edwards and Van Tongeren [62] demonstrated the mediating effect of meaning in life on the link between suffering and well-being. Heisel and Neufeld [63] showed that a sense of meaning mediates the relationship between the reasons for living and suicidal ideation among community-dwelling older adults. This mediating effect can be explained by Frankl’s theory [49]. Frankl (in 1959) suggested three pathways for finding meaning in life: creative, experiential, and attitudinal [49]. Among these, the experiential pathway involves finding meaning by encountering individuals, arts, and nature. The creative pathway pertains to finding meaning in life by doing something (e.g., work or deeds). Lonely older adults who perceive disconnectedness from individuals, activities, and work, are more likely to restrict opportunities to find meaning in life through experiential and creative pathways. In other words, older adults who experience a high level of perceived loneliness may be less likely to be involved in activities, social interactions, and behavioral routines, which could negatively impact finding meaning in life and result in the development of depressive symptoms. 

Notably, our findings from the serial mediation model illustrated an intermediate line in the path of loneliness -> search for meaning -> presence of meaning -> depressive symptoms. Thus, H4 was supported. The search for meaning did not directly relate to depressive symptoms among older adults; it was linked to the presence of meaning. This serial mediating effect of the search for and presence of meaning is supported by the “search-to-presence” model, which suggests that the search for meaning precedes the experience of having meaning in life [60]. Thus, individuals who actively seek meaning are more likely to discover meaning in their lives [60]. Previous studies have reported that a high level of the search for meaning is linked to a high level of the presence of meaning [32,64,65,66]. Cultural factors also influence this relationship. Some studies conducted in Eastern cultures, including South Korea, have consistently reported that the search for meaning is positively linked to the presence of meaning [32,67,68,69]. Conversely, studies conducted in Western cultures have shown that the search for meaning is negatively associated with the presence or absence of meaning [30,60,67,70,71]. This is attributed to the fact that Western individuals tend to perceive the search for meaning and presence of meaning as separate components, considering that pursuing meaning in life does not necessarily lead to the discovery of meaning. In contrast, Eastern cultures, which exhibit more holistic cognitive characteristics, tend to perceive the search for meaning and presence of meaning as interconnected components of a single process [67]. In summary, our findings suggest that merely searching for meaning is not sufficient to positively impact relationships among older adults. To prevent depressive symptoms, lonely older adults who search for meaning in life need help in experiencing meaning in their lives. 

This study had some limitations. First, data were obtained from participants using convenience sampling. Therefore, caution should be exercised when generalizing these findings to all older adults in South Korea. Our cross-sectional design was also a limiting factor. The relationship between loneliness, meaning in life, and depression may be bidirectional. Previous studies have reported bidirectional relationships between loneliness and depressive symptoms [72,73,74,75,76], and between loneliness and meaning in life [29,77,78]. Therefore, longitudinal research designs are required to address bidirectional relationship issues between the study variables. Even though we accounted for demographic variables, we did not take into account the impact of the pandemic. This external factor should be taken into account when interpreting the results. Moreover, potential confounding factors, such as economic status and cognitive impairment, should be assessed and controlled in future studies. Finally, further research is needed to investigate the mechanism whereby searching for meaning leads to finding meaning in life among older adults, while considering the cultural context. 

Despite these limitations, this study extends our understanding of the mechanisms underlying the relationship between loneliness and depressive symptoms in older adults. Our findings suggest that a sense of meaning in life plays a significant role in suppressing depressive symptoms in older adults who feel lonely. It is also essential to help older adults who search for meaning achieve a sense of meaning in their lives to prevent depressive symptoms. Considering the loss of social networks owing to retirement, lost spouses, and close acquaintances later in life, loneliness cannot be easily addressed among older adults. However, meaning and purpose in life already exist to be found [21]; thus, meaning in life could be enhanced by effective interventions to discover it [79]. Specifically, interventions that assist lonely older adults in continuing their pursuit of meaning through activities, social interactions, and behavioral routines could help prevent the development of depression. Pursuing meaning in life can be found in modest, everyday activities, including routine tasks, physical activities, and helping others in small ways. Additionally, the findings of this study reveal that it is not only important for the older adults to engage in activities in the pursuit of meaning but it is also crucial that they recognize and find meaning in these activities, experiencing their lives as sufficiently meaningful and purposeful.

## 5. Conclusions

This study investigated the relationship between loneliness and depressive symptoms, and the serial mediating effects of the search for meaning and the presence of meaning among older Korean adults during the pandemic. In addition to efforts to directly reduce loneliness, our findings emphasize the importance of meaning in life among lonely older adults. Our findings could be beneficial in guiding future prevention and intervention efforts to address depressive symptoms in lonely older adults. Maintaining and enhancing meaning in life is valuable for older adults, especially in crises such as the COVID-19-induced pandemic. 

## Figures and Tables

**Figure 1 healthcare-12-00584-f001:**
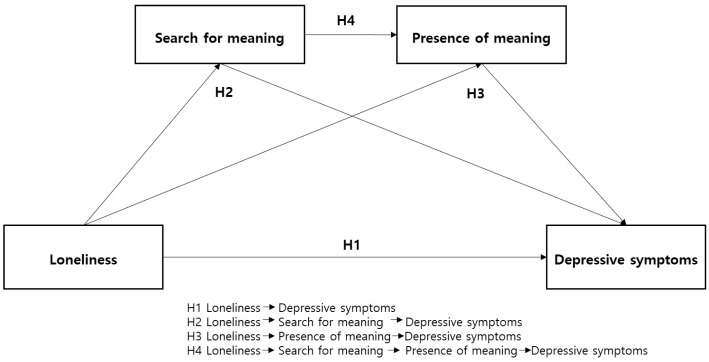
Hypothesized model of the study.

**Figure 2 healthcare-12-00584-f002:**
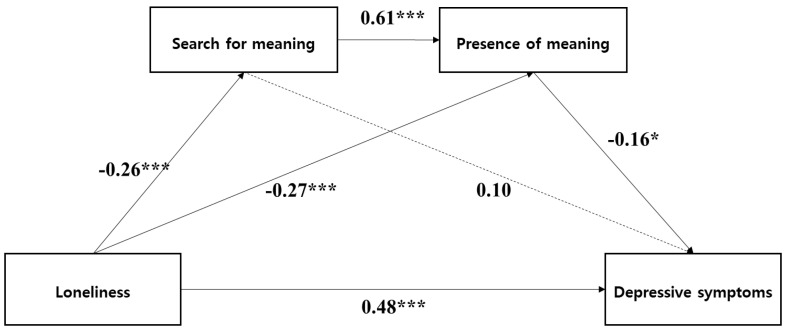
Mediation model with standardized path coefficients. Note: For simplicity, controlling variables and factor loadings are not displayed for the latent variables. *** <0.001, * <0.05.

**Table 1 healthcare-12-00584-t001:** Characteristics of study participants (N = 213).

Variables	M ± SD/n(%)
Age (year)	75.07 ± 6.60
Gender	
Male	85 (39.9)
Female	128 (60.1)
Having spouse	
No	85 (39.9)
Yes	128 (60.1)
Education level	
Elementary school	78 (36.6)
Middle school	44 (20.7)
High school	67 (31.5)
College or above	24 (11.3)
Religion	
No	60 (28.2)
Yes	153 (71.8)
Perceived health status (range 3~15)	9.21 ± 2.05

**Table 2 healthcare-12-00584-t002:** Descriptive statistics and correlation analysis of the study variables.

	Mean	SD ^1^	Skewness	Kurtosis	Loneliness	Search for Meaning	Presence of Meaning	Depressive Symptoms
Loneliness	1.86	0.48	0.25	0.14	-			
Search for meaning	5.04	1.22	−0.83	0.33	−0.240 (<0.001)	-		
Presence of meaning	4.94	1.11	−0.51	0.03	−0.409 (<0.001)	0.705 (<0.001)	-	
Depressive symptoms	7.06	5.51	1.05	1.12	0.529 (<0.001)	−0.178 (0.009)	−0.356 (<0.001)	-

^1^ SD: Standard Deviation.

**Table 3 healthcare-12-00584-t003:** The results of the mediating effect test.

Paths	Coefficient	Lower Bound	Upper Bound	*p*-Value
Loneliness > CESD (Direct effect)	0.520	0.413	0.627	<0.001
Loneliness > MLQ-S > CESD	−0.027	−0.079	0.008	0.123
Loneliness > MLQ-P > CESD	0.043	0.003	0.095	0.039
Loneliness > MLQ-S > MLQ-P > CESD	0.025	0.003	0.063	0.032

## Data Availability

The data presented in this study are available on request from the corresponding author.

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
