# Peer review of "“Meaning in Life” Mediates the Relationship between Loneliness and Depressive Symptoms among Older Adults during the COVID-19 Pandemic"

_healthcare, 2024, doi:10.3390/healthcare12050584_

Round 1

Reviewer 1 Report

Comments and Suggestions for Authors

Dear Authors,

good work but I think (personally) that the choise of "Hypothesized of the study" may be not fully valid or at least questionable. There could not be a bidirectional relationship and between variables themeselves to influence the loniless? I suggest to added (eventually) the point in the limit (I hope of having been clear). The choise to not include the conclusion and not divide discussion and conclusion is questionable; I suggest to structure of two distinct session.

Best regards

Author Response

We would like to express our appreciation for your extremely thoughtful comments.

Thanks to the comments you gave us, we were encouraged to study. Your feedback was extremely helpful to our article.

Thank you again for taking the time to share your feedback.

Sincerely yours,

The authors

Reviewer 2 Report

Comments and Suggestions for Authors

Upon reviewing the manuscript, I found it to be well written. However, there are a few areas that warrant the authors' attention.

While you aptly discuss the significance of the meaning of life in general, there is a need to delve into its relevance specifically for the older population, given the pronounced influence of age on life's perspectives. Hence, further elaboration on the meaning of life for older individuals is warranted.

Could you provide an elucidation of Hypothesis 4 depicted in the figure? Additionally, it would be beneficial to include all hypotheses within the figure for clarity.

How did you account for external confounders such as the pandemic, religiosity which may impact the perception of meaning in life among the older population? Please elaborate on the methods employed to calculate the necessary sample size to fulfill the study objectives, as well as the sampling methodology utilized to ensure the generalizability of the results to the Korean older population. Furthermore, what measures were taken to assess the reliability of the Korean versions of all the scales used in the study? Additionally, please elucidate on the techniques employed to control for covariates in the analysis.

Could you elaborate on how perceived health status was defined in Table 1 and the methodology employed to establish its range?

While the analysis indicates a significant correlation in Table 2, the strength of the correlation appears to be modest. Could you provide insights into this observation?

The limitations section should elucidate on potential confounding factors inherent in the study. Additionally, the recommendations should underscore the practical implications of the study in guiding optimal treatments for the older population.

Minor Concerns:

Abstract: Remove numbering.

Introduction: Line 42 - Ensure consistency in citations and some other parts in the discussion.

Certainly, including a conclusion, even if brief, can enhance the overall structure and completeness of the manuscript. However, it's essential that the conclusion adds value by summarizing key findings and insights from the study. If there isn't significant new information to add or if the conclusion would merely repeat what has already been stated in the discussion or results sections, then it may be omitted.

These revisions aim to enhance the scientific rigor and clarity of the provided feedback.

Author Response

(The authors gave the same response as above.)

Reviewer 3 Report

Comments and Suggestions for Authors

Dear Authors,

The present research article, entitled ““Meaning in life” mediates the relationship between loneliness 2 and depressive symptoms among older adults during the 3 COVID-19 pandemic”, aims to examine the relationship between loneliness and depressive symptoms, whether this relationship can be mediated by meaning in life, and the differential role of the search for and presence of meaning in relation to depressive symptoms among older adults.

The main strength of this study is that it aims to examine the mediating role of meaning seeking and meaning presence in the relationship between loneliness and depressive symptoms among older adults. Determining this aspect can help to implement new approaches to intervening with this population.

In general, I believe that the topic and approach of this article is timely and of interest to the readers of Healthcare. However, I believe that some issues should be included to improve the quality of the manuscript.

Abstract:

·         There is no background information, only the target, do you think it is possible to include anything else?

·         The conclusions are similar to the results. I recommend discussing the implications of the results.

      Materials and Methods

·         What do you mean by “community-dwelling” older adults? I think it is better to clarify this or use another term.

·         If the study was conducted in the pandemic, how were the participants assessed? Online or in a traditional pen and paper manner? This would be good to explain.

·         How long did the assessment last?

 Best regards.

Author Response

(The authors gave the same response as above.)
